# CONTEXTSPEECH: EXPRESSIVE AND EFFICIENT TEXT-TO-SPEECH FOR PARAGRAPH READING

## ABSTRACT

Although Text-to-Speech (TTS) has made rapid progress in speech quality at sentence level, it still faces a lot of challenges in paragraph / long-form reading. Synthesizing sentence by sentence in a paragraph and then concatenating them together will cause inconsistent issues that affect paragraph-level expressiveness. While directly modelling all the sentences in a paragraph will incur large computation / memory cost. In this paper, we develop a TTS system called ContextSpeech, which models the contextual information in a paragraph for coherence and expressiveness without largely increasing the computation or memory cost. On the one hand, we introduce a memory-cached recurrence mechanism to let the current sentence see more history information both on the text and speech sides. On the other hand, we construct text-based semantic information in a hierarchical structure, which can broaden the horizon and incorporate the future information. Additionally, we use a linearized self-attention with compatible relative-position encoding to reduce the computation / memory cost. Experiments show that ContextSpeech significantly improves the paragraph-level voice quality and prosody expressiveness in terms of both subjective and objective evaluation metrics. Furthermore, ContextSpeech achieves better model efficiency in both training and inference stage.

## 1 INTRODUCTION

Deep neural network based Text-to-speech (TTS) develops rapidly and brings significant breakthroughs. Neural Network based models like Tacotron 1/2 Wang et al. (2017); Shen et al. (2018), Deep voice 3 Ping et al. (2017), Transformer TTS Li et al. (2019), FastSpeech1/2 Ren et al. (2019; 2020), together with neural vocoder model like WaveNet Oord et al. (2016), Parallel WaveNet Oord et al. (2018), WaveRNN Kalchbrenner et al. (2018), MelGAN Kumar et al. (2019) can generate high-quality voices. More recently it evolves to fully end-to-end model like VITS Kim et al. (2022) and NatualSpeech Tan et al. (2022), which achieves high-fidelity and close-to-recording quality.

These TTS models usually convert text to speech in sentence-level. In fact, there are many scenarios in TTS where the synthesized audio is created in paragraph-level, like news reading, audiobook, audio content dubbing, or even dialogue composed by multiple interrelated sentences. Regarding the large variation of context in long-form content, concatenating synthesized speech sentence by sentence still has obvious gap to natural recording in paragraph reading from perceptual evaluation. Meanwhile, the imbalanced distribution of TTS corpus data with long-tail sentences, *e.g.*, extra-long or extra-short sentences, makes it difficult for TTS systems to generate high quality synthesized speech in such context. From our observation, the sentence-level speech synthesis is limited as followings:

- Correlation between adjacent sentences. For paragraph reading, adjacent sentences influence each other naturally as the semantic information flowing. Thus, sentence-level synthesis lacks coherence between adjacent sentences and accurate expression.

- Efficiency or consistency on extra-long sentences. Synthesizing extra-long sentences usually leads to unstable results (e.g. bad alignment between text and speech) and high latency. Generally, such sentences will be cut into two or more segments and then synthesized separately, which may cause inconsistent speech rate or prosody between cut segments.

- Quality on extra-short sentences. As rare patterns in corpus, sentences that are too short (e.g. consisted by one or two words) are easy to generate audio with poor quality (e.g. bad pronunciation or extremely slow speech rate).

Naturally, involving contextual-related information should be helpful to address above issues and do better paragraph-level text-to-speech. To be more specific, 1) for adjacent sentences, contextual information brings inter-sentence knowledge, to guide the generation of current sentence under more appropriate speaking style. 2) For extra-long sentences, intra-sentence contextual information can benefit each position to generate expressive and consistent audio. 3) For extra-short sentences, leveraging contextual information can enlarge the perceptive scope to produce a more stable result.

Nevertheless, contextual information is difficult to define and the introduction of external information into the model leads to high calculation and memory cost. Thus, the challenges of this work are 1) how to construct and leverage effective contextual information; and 2) how to reduce the cost introduced by the external information. Based on such motivation, we propose the ContextSpeech model, and the mainly contributions of this work are summarized as follows:

- We present a framework with cached hidden state to capture previous information, thus it evolves from sentence-level to paragraph-level speech synthesis in perspective of modeling. We use one of the state-of-the-art sentence-level speech synthesis architecture, Conformer Peng et al. (2021) based TTS in Liu et al. (2021), as our baseline framework. This model can capture local correlation well and produce more expressive speech compared with Feed-Forward Transformer based FastSpeech. Based on that, referring to segment-level recurrence mechanism proposed in Dai et al. (2019), the cached hidden state for each Conformer block in both encoder and decoder transfers text and speech information from the previous segment to current segment.

- Inspired by the context-aware conversational TTS Guo et al. (2021), we propose a modified version of text-based contextual encoder and integrate it into our memory reuse mechanism based conformer model. We first extract BERT Devlin et al. (2018)-based embedding and pre-defined statistical information from contextual scripts. After that, we build a text-based contextual encoder to consume these features and then combine them with phoneme embedding to go through the Conformer block. Such integration of the context information can broaden the model horizon from history to future and alleviate the one-to-many mapping issue in TTS Ren et al. (2020).

- To reduce the memory and calculation cost, we use a linearized self-attention module to avoid quadratic complexity caused by softmax self-attention. Meanwhile, we adopt a permute-based relative position encoding to fit the efficient self-attention under our memory reused framework.

We conduct experiments on a speech corpus of Chinese audiobook. The results show that ContextSpeech can generate more expressive and coherent paragraph audios compared with baseline ConformerTTS model in terms of objective and subjective evaluation. From the observation, it also alleviates the issues caused by extra-long and extra-short sentences obviously. And the ablation experiments demonstrate that both of our proposed contextual model framework and the integration of text-based contextual encoder are effective. Additionally, the efficiency optimization on ContextSpeech expand around 2x of both memory tolerance and training speedup, and the final model largely alleviate the efficiency issue of extra-long input compared with baseline model.

## 2 RELATED WORK

### 2.1 TEXT TO SPEECH

TTS is a technique aimed at converting given text to speech automatically, and the goal of which includes naturalness, expressiveness, robustness and efficiency. Along with the flourishing of deep learning, the conventional methods like concatenative synthesis Hunt & Black (1996) and statistical parametric synthesis Zen et al. (2009) are gradually replaced by neural network based approaches Tan et al. (2021). Autoregressive acoustic model Wang et al. (2017); Shen et al. (2018); Ping et al. (2017); Li et al. (2019) and vocoder Oord et al. (2016); Kalchbrenner et al. (2018) largely improves the

voice quality but are limited by the slow inference speed, especially on the vocoder part. To improve the model efficiency, non-autoregressive vocoder methods Oord et al. (2018); Prenger et al. (2019); Kumar et al. (2019) are proposed for boosting the inference speed and move the bottleneck to acoustic model. Therefore, non-autoregressive acoustic models Ren et al. (2019; 2020); Kim et al. (2020); Łańcucki (2021) appeared. For example, FastSpeech Ren et al. (2019) is one of the most remarkable work that is fast and stable based on the duration predictor and feed-forward transformer in both encoder and decoder. But the expressiveness is limited due to teacher-student distillation (information loss). An improved Conformer Hunt & Black (1996)-based FastSpeech model is proposed in Liu et al. (2021), we call it ConformerTTS. Combining convolutional neural networks and multi-head self-attention for both local and global information encoding, this model can generate audios with high naturalness and expressiveness. In this paper, we will use the ConformerTTS as our baseline model, and then make it context-aware to improve the quality and efficiency on paragraphs and extremely short/long sentences.

## 2.2 Context Modeling

Contextual information can be leveraged from two perspectives, priori knowledge and model construction. Text-based information processing is actually an indispensable basement. Previously, front-end processing is an essential module in a classical SPSS (Statistical Parametric Speech Synthesis) system, which is used for converting text to linguistic features. This module is replaced by a powerful encoder with phoneme or character as input in current end-to-end neural TTS model. Even so, incorporating external linguistic and semantic information can still benefit the voice quality Xiao et al. (2020); Guo et al. (2021); Lei et al. (2022); Xu et al. (2021); Xue et al. (2022). On the other hand, variant model structures serve to extracting contextual information. For example, recurrent mechanisms Hochreiter & Schmidhuber (1997); Chung et al. (2014); Wang et al. (2017); Shen et al. (2018) are good at modelling sequence but limited to sequential computation. Convolutional block Kalchbrenner et al. (2016); Gehring et al. (2017); Ping et al. (2017); Arık et al. (2017) can capture more detail information but lack long-range dependencies. Transformer-based models Vaswani et al. (2017); Li et al. (2019) can achieve better scalability and model global correlation information well. These models mainly focus on the modelling of intra-sequence correlation. For the cross-sentence context modeling, Transformer-XL Dai et al. (2019) is proposed to deal with inter-segment information transfer based on a segment-level recurrence mechanism, and similar framework is introduced into TTS task Wang et al. (2020b). In this paper, we capture fine-grained and global correlation by convolutional based Transformer framework, learn dependency of adjacent sequences with XL-like hidden state reuse and construct a text-based contextual encoder to incorporate both linguistic and semantic information.

## 2.3 Efficient Transformer

Transformer model achieves success across a wide range of domains. The self-attention module brings the effectiveness but also limits the model efficiency due to the quadratic time and memory complexity. Especially for extra-long sequences, the workload will be heavy and largely hinder model scalability. Recently, a number of Transformer variants with different mechanisms are proposed to address this efficiency issue Tay et al. (2020). For instance, Sparse Transformers Child et al. (2019) compute a sparse number of query-key pairs based on fixed attention patterns. Reformer Kitaev et al. (2020) introduces Locality Sensitive Hashing to measure the similarity of tokens efficiently. Linformer Wang et al. (2020a) improves efficiency by assuming the attention matrix as low-rank structure to be decomposed. In this paper, as we will leverage external contextual information and structures, to make the model more efficient, we use the linearized attention from Linear Transfomer Katharopoulos et al. (2020). It is a kernel based method that can significantly reduce the computation time and memory footprint even linearly according to the context length, which is also demonstrated in TTS scenarios Xiao et al. (2022).

## 3 ContextSpeech

In this section, we will introduce detailed architecture of our proposed model, ContextSpeech. In Section 3.1, we first describe our baseline framework, ConformerTTS, which is a Conformer based FastSpeech model that can capture sentence-level local and global information well. Based on that, we introduce the memory reuse mechanism, which can transfer contextual information of both text and

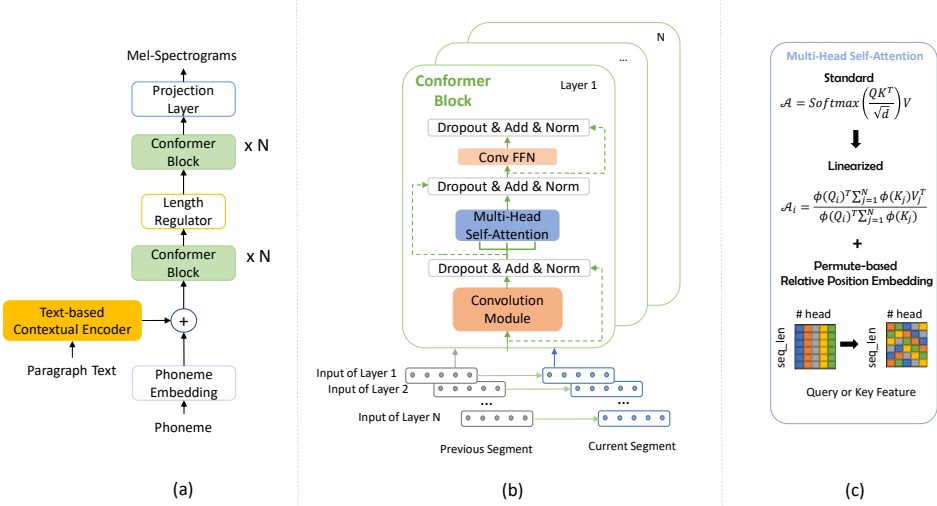

Figure 1: Overall Model Architecture of ContextSpeech. (a) ConformerTTS Framework (b) Conformer block with Segment-Level Memory Reuse (c) Linearized Self-Attention with Permute-based Relative Position Embedding.

speech between adjacent sentences. In Section 3.2, we illustrate the proposed text-based contextual encoder, which can encode linguistic and semantic information effectively in a hierarchical structure. Finally, Section 3.3 presents the efficient self-attention mechanism we used in our framework, which significantly improves the model efficiency in both training and inference stage.

## 3.1 CONFORMERTTS WITH MEMORY REUSE

**ConformerTTS (Baseline)**. Figure 1-(a) is the overall framework of a ConformerTTS model (without the text-based contextual encoder). The difference between it and FastSpeech is that the Feed-Forward Transformer (FFT) block is replaced by the Conformer Block(CB) Hunt & Black (1996) in both encoder and decoder Liu et al. (2021). As shown in Figure 1-(b), the CB integrates a Convolution Module (ConvM) and a Multi-Head Self-Attention (MHSA) to model the local correlation and the global interaction. Additionally, a convolution based Feed-Forward Network (ConvFFN) is attached after the self-attention for encoding the correlation between adjacent hidden state in further. To be specific, the ConvM is composed of four stacked components, including a convolutional feed-forward module, a gated linear unit (GLU), a depthwise convolution module and another convolutional feed-forward module. Set N as the number of CB stacked in encoder (or decoder), the input feature of the n-th CB can be represented as $H_t^n = [h_{t,1}, ..., h_{t,L}]$, where $t$ is the index of current sequence and $L$ is the sequence length. This model will be used as baseline framework in this paper.

**Segment-level Memory Reuse**. Refer to Transformer-XL Dai et al. (2019), we cache the hidden state of previous segment in each layer and reuse it with current segment for involving contextual information as shown in Figure 1-(b). Notice that, the previous segment is with a fixed length while we use a complete sentence as our current segment to retain more intact semantic and acoustic information from both text and speech. Instead of reusing the input feature of MHSA, we choose to cache the input feature of CB directly since the ConvM can help in capturing the contextual information around the concatenation point. As the output of the $n$-th block is the input of the $(n + 1)$-th block when $n < N$, the hidden state can be represented as Eq.(1), where $SG(\cdot)$ means stop-gradient and the notation $[A \circ B]$ indicates concatenating hidden sequences $A$ and $B$ along the length dimension.

$$H_t^{n+1} = [SG(H_{t-1}^{n+1}) \circ ConformerBlock(H_t^n)] \tag{1}$$

## 3.2 TEXT-BASED CONTEXTUAL ENCODER

Given the same sentence with different context, the prosody of the generated speech would be different. In this section, we adopt a text-based contextual encoder to enhance the prosody expressiveness and

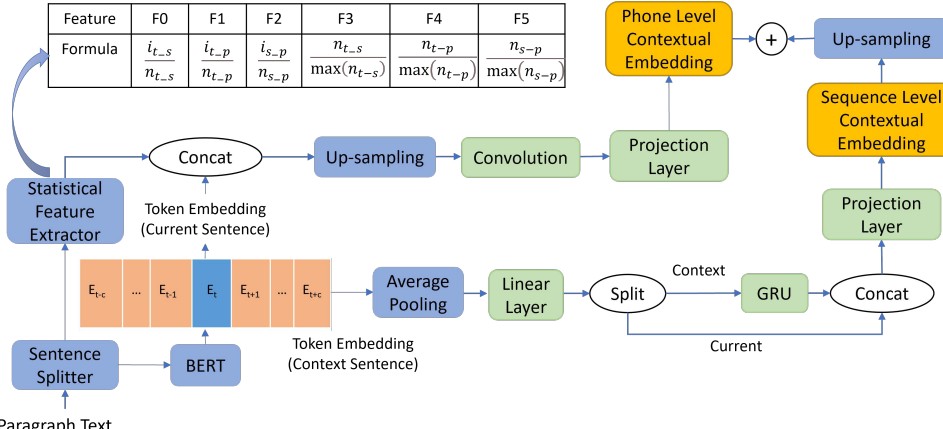

Figure 2: Text-based Contextual Encoder.

coherence. Overall, we use two modules to extract semantic and syntax information separately, then combine these features together.

As shown in Figure 2, given a paragraph text, a sentence splitter firstly splits the paragraph text into sentences. Then, for each sentence centered in a predefined range of context $c$, we use a BERT Devlin et al. (2018) model to extract the token [1]-level semantic context embedding derived as current sentence token embedding $E_t$ and context sentence token embedding $E_{t-c:t+c}$ for further operation. At the same time, a token-level statistical feature extractor is used to calculate the syntactic information, which covers the correlation from sentence to paragraph. The calculated statistical features are listed in the upper table of Figure 2, where $i_{t\_s}$ means the $i$-th token in sentence, $i_{t\_p}$ means the $i$-th token in paragraph, $i_{s\_p}$ means the $i$-th sentence in paragraph, $n_{t\_s}$ means the number tokens in a sentence, $n_{t\_p}$ means the number of tokens in a paragraph, $n_{s\_p}$ means the number of sentences in a paragraph. The maximum values are used for normalization and are counted based on the training data.

- Phoneme-level contextual embedding. When BERT and statistical features of current sentence are ready, they will be concatenated and then up-sampled to align with the phoneme sequence. After that, a convolution layer and projection layer are used to generate the phoneme-level context-aware feature.

- Sequence-level contextual embedding. BERT token embedding for the predefined range of context in the paragraph will go through the average pooling. After a linear layer, the context embedding will be encoded by a GRU module to a state vector, which is the paragraph-level contextual representation and contains both historical and future information. Then, the paragraph-level contextual representation will be concatenated with the current sentence's embedding and go through a projection layer to get the sequence-level context-aware feature.

Our text-based contextual encoder contributes in two aspects: (i) it broadens the horizon of current phoneme to paragraph scope by calculating the paragraph-level statistical features and generating the higher-level contextual state vector, rather than just using the current utterance. (ii) the constructed hierarchical contextual feature will be inserted into phoneme embedding which impacts encoder effectively.

### 3.3 EFFICIENT SELF-ATTENTION MECHANISM

**Linearized Self-Attention**. Let $X \in \mathbb{R}^{L \times d}$ as the input of self-attention module, $Q = W_q \cdot X$, $K = W_k \cdot X$ and $V = W_v \cdot X$ are linear transformations on the input $X$. The canonical softmax-based self-attention mechanism in Transformer can be presented as equation (2), where the time and memory complexity is quadratic according to the input length.

$$\mathcal{A}(Q, K, V) = softmax(QK^T / \sqrt{d})V \tag{2}$$

---

[1]For Chinese, "token" means "character", for English, "token" means "subword".

$$\mathcal{A}(Q_i, K, V) = \Big( \sum_{j=1}^{L} sim(Q_i, K_j)V_j \Big) / \Big( \sum_{j=1}^{L} sim(Q_i, K_j) \Big) \tag{3}$$

As in Katharopoulos et al. (2020), the attention matrix can be generalized as a similarity function of $Q_i$ and $K_j$, the $i$-th or $j$-th row of the matrix $Q$ and $K$, as Eq.(3). The similarity function can be any other attention functions that are non-negative. Given a qualified kernel function $\phi(x)$, the generalized row-wise attention matrix can be rewritten as Eq.(4).

$$\mathcal{A}(Q_i, K, V) = \Big( \sum_{j=1}^{L} \phi(Q_i)^T \phi(K_j)V_j \Big) / \Big( \sum_{j=1}^{L} \phi(Q_i)^T \phi(K_j) \Big) \tag{4}$$

$$= \Big( \phi(Q_i)^T \sum_{j=1}^{L} \phi(K_j)V_j \Big) / \Big( \phi(Q_i)^T \sum_{j=1}^{L} \phi(K_j) \Big) \tag{5}$$

According to the associative property of matrix multiplication, $\phi(Q_i)^T$ can be taken out of the summation formula both in numerator and denominator as Eq.(5). Thus, we can compute the summation formula part in advance and reuse them for each query.

**Permute-based Relative Position Encoding**. Since we reuse the hidden states as described in Section 3.1, we have to make sure the positional information carried by memory is consistent with that of current segment. Transformer-XL propose a relative positional encoding method while it is target for softmax-based self-attention. Since we use a linearized self-attention instead, we need a corresponding compatible relative-positional encoding way. Therefore, we applied the permute-based method proposed in Chen (2021), where the $sim(Q_i, K_j)$ in Eq.(3) will be converted to permute based format as Eq.(6). $r$ is set as 1 to avoid exploding as the sequence length increases. A premutation $B$: $\{1,2,...,d\}$ -> $\{1,2,...,d\}$ is generated randomly, where $d$ is the dimension of query or key. Here, the first $\{1,2,...,d\}$ and the second $\{1,2,...,d\}$ can be treated as index collections with different order. $P_B$ is the corresponding premutation matrix of $B$, where $P_{B,ij} = 1$ if $B(i) = j$; otherwise $P_{B,ij} = 0$.

$$sim_p(Q_i, K_j) = \Big( r_i P_B^i \phi(Q_i) \Big)^T \Big( r^{-j} P_B^j \phi(K_j) \Big) \tag{6}$$

## 4 EVALUATION

### 4.1 EXPERIMENTAL SETUP

**Dataset**. We did experiments on an expressive Chinese male voice. The dataset is an audiobook corpus which is composed of around 70 hours ($\sim 35,000$ sentences) of narration speech data and the corresponding text transcripts. We left 100 paragraphs from the same book for objective evaluation and construct 3 different paragraph test sets from other books for subjective evaluation. These 3 test sets includes: Set-A: 50 paragraphs with sentences in normal length, in order to evaluate the overall model performance on paragraph reading. Set-B: 50 paragraphs with sentences in extra-short length (e.g. one or two words), to see if the model relieve the robustness issue in extra-short sentences. Set-C: 10 paragraphs with incremental sentence number from 2 to 11, which is used for testing the model efficiency of extra-long input.

**Model Configuration**. Our ContexSpeech model is composed by encoder (for phoneme side), variance adapter (pitch and duration predictor) and decoder (for mel-spectrograms side). There are 6 stacked Conformer blocks in both encoder and decoder and 4 self-attention heads in each Conformer. In Convolution Module, the [input_dim, output_dim, kernel_size] for the first pointwise convolution module, the depthwise convolution module and the second pointwise convolution module are [384, 768, 1], [384, 384, 7] and [384, 384, 1] respectively. The kernel function used in linearized self-attention is $\phi(x) = elu(x) + 1$, and the ConvFFN consists of 2 convolution layers with input and output dimension as 384 and hidden state dimension as 1536. The memory length is set as 128 for encoder, which is around the average input length of one sentence in the training data. While the memory length of decoder is set as 64, which is the setting with best objective evaluation results among the setting group [16, 32, 64, 128, 256, 512]. The experimental result of memory length selection is presented in Appendix B. For the text-based contextual encoder, the pretrained Chinese

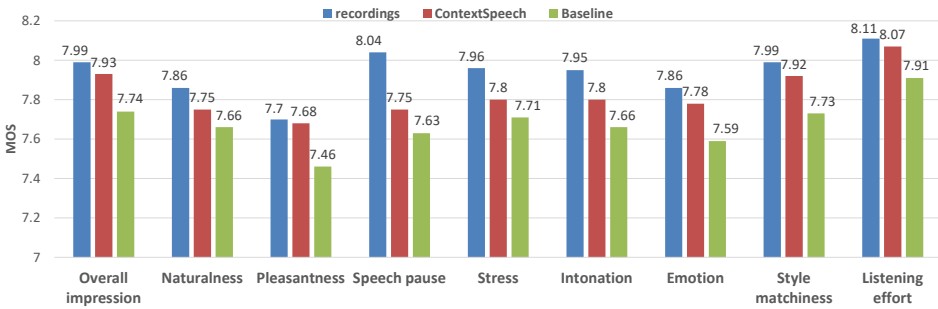

Figure 3: Paragraph MOS result on Recordings, Baseline, ContextSpeech.

Table 1: Objective Metrics Result of Baseline and ContextSpeech model.

| Metrics | Correlation | | | | RMSE | | | |
|---|---|---|---|---|---|---|---|---|
| | Pitch | Intensity | Duration | Pause | Pitch | Intensity | Duration | Pause |
| Baseline | 0.688 | 0.853 | 0.764 | 0.888 | 25.398 | 11.582 | 47.090 | 60.803 |
| ContextSpeech | 0.716 | 0.870 | 0.817 | 0.929 | 23.197 | 10.929 | 44.658 | 36.145 |

BERT-based model Wolf et al. (2019) is used for our experiments, the size of the extracted BERT embedding and the statistical features are 768-dim and 6-dim, respectively. For the phone level contextual extractor, the [input_dim, output_dim, kernel_size] of the convolution layer is [774, 384, 5], which is followed by RELU, layer norm, dropout with rate 0.5 and a linear layer with size [384, 384]. For the sequence level contextual extractor, the size of linear layer is [768, 384], and the context size in our experiment is 11, i.e., 5 sentences before and after except the current sentence. The GRU layer with size [384, 384] is used to encode the input context embedding to a state vector, and followed by a RELU, dropout with rate 0.5 and a linear layer with size [768, 384].

**Evaluation**. We conduct paragraph MOS (mean opinion score) test to evaluate the overall voice performance of our method considering both recording and baseline model. In the paragraph MOS test, 25 native speakers listen to each audio and give a score in 10-point scale according to the overall performance and each specific metric, including naturalness, pleasantness, speech pause, stress, intonation, emotion, style matchiness and listening effort (Definition of each metrics is given in Appendix C). Paragraph CMOS (comparative mean opinion score) test is used to compare the proposed model with the baseline model in different test sets. In the paragraph CMOS test, given the same text, 15 native speakers will compare the synthesized samples side by side and give a score from -3 to +3, where the baseline model is set as 0 for reference. Additionally, we propose a group of objective metrics to evaluate model performance according to recordings with the same transcripts, including pitch, intensity, duration and pause. Detailed definition and calculation are listed in Appendix A. Objective metrics are useful in model parameter and final step selection since we need to compare many groups of samples, which is heavy for subjective evaluation. For model efficiency evaluation, we do model training on 8 NVIDIA V100 GPUs and inference work on 1 NVIDIA Tesla K80 GPU.

## 4.2 QUALITY ON PARAGRAPH READING

We conduct a paragraph MOS test on Set-A for ContextSpeech model along with baseline and recording. Figure 3 shows the result in overall impression and other 8 specific metrics. The ContextSpeech outperforms baseline model no matter in which metric and is close to recording in overall impression (7.93@7.99). To be more specific, the proposed model reduces the MOS gap with recording from 0.25 to 0.06 compared with baseline model, which is around 76% reduction. Especially for voice pleasantness, emotion, style matchiness and listening effort, ContextSpeech model shows significant improvement with more than 50% MOS gap reduction.

Besides the subjective evaluation, we also calculate the prosody-related objective metrics to measure the similarity between synthesized voice and 100 paragraph recordings. Table 1 shows that ContextSpeech achieves improvement in each objective metric compared with baseline model, which also verifies the model performance in paragraph-level prosody expressiveness.

Table 2: CMOS result on Paragraphs with Extra-Short or Long sentences.

|  | Baseline | ContextSpeech |
|---|---|---|
| Paragraphs with Extra-short Sentences | 0 | +0.107 |
| Paragraphs with Extra-long Sentences | 0 | +0.226 |

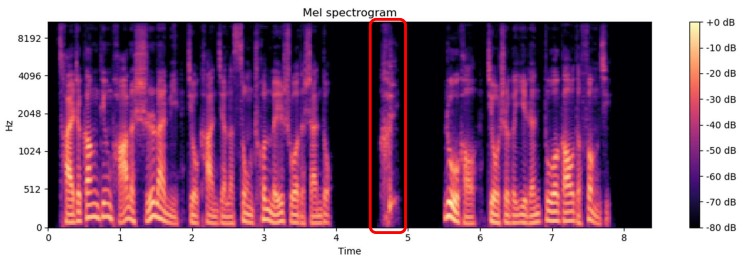

(a) Mel-Spectrograms of Baseline sample

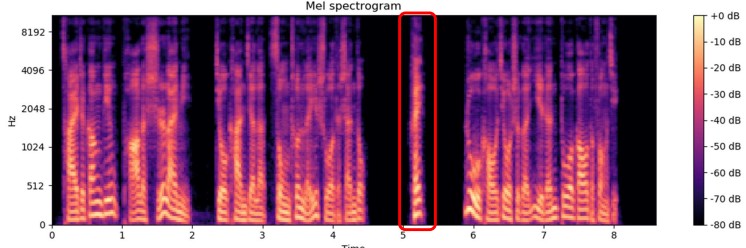

(b) Mel-Spectrograms of ContextSpeech sample

Figure 4: Mel-Spectrogram samples of paragraph with one-word sentence

### 4.3 STABILITY ON EXTRA-SHORT SENTENCE

As mentioned in Section 1, extra-short sentences (one or two words) in sentence-level speech synthesis model usually cause robustness issue, such as bad pronunciation and low speech rate. Therefore, we conduct paragraph CMOS test on Set-B. Set score of Baseline model as 0 for reference, ContextSpeech obtains 0.107 CMOS gain. Two kinds of issues on extra-short sentence are obviously relieved, including bad pronunciation issue in one-word sentences and low speech rate issue in two-word sentences. Figure 4 demonstrates the mel-spectrogram samples to compare baseline and ContextSpeech model in terms of one-word sentence. The red rectangles mark the position of the sentence with only one word in the paragraph. It is obvious that the spectrogram of baseline model in that position is muffle (Figure 4(a)), while that of ContextSpeech model is much clearer with complete formant (Figure 4(b)). By listening to the audios, we can also notice that the pronunciation of that one-word sentence is distorted in baseline paragraph but clear in ContextSpeech paragraph. The sample pairs with pronunciation and speech rate issues will be attached in Appendix D.

### 4.4 EFFICIENCY ON EXTRA-LONG SENTENCE

The efficient self-attention module described in Section 3.3 largely improves the model efficiency. For training stage, ContexSpeech with linearized self-attention can achieved 2x of speedup and 2x of memory tolerance compared with using softmax-based self-attention. For Inference stage, ContextSpeech shows significant advantage over the baseline model especially for extra-long input. We construct a test consisting of 10 paragraphs, and the sentence number is incremental from 2 to 11. Table 3 illustrates the inference latency for baseline and ContextSpeech model according to different input phoneme length on 1 NIVIDA K80 GPU. The baseline model run into out-of-memory when the input phone number increase to 1574, while the ContextSpeech model can still handle such long sequence. Additionally, the ContextSpeech model outperforms the baseline model in each group and achieves more than 10x speedup compared with baseline model when the input length is 1506. Furthermore, we do paragraph CMOS on this test set and obtain 0.226 CMOS gain (Baseline vs

Table 3: Inference latency – millisecond per phone in different length of input.

| #Sent(#Phone) | 2(252) | 3(414) | 4(506) | 5(620) | 6(780) | 7(898) | 8(1194) | 9(1354) | 10(1506) | 11(1574) |
|---|---|---|---|---|---|---|---|---|---|---|
| Baseline | 1.116 | 0.521 | 0.474 | 0.422 | 0.403 | 0.416 | 0.720 | 0.609 | 0.797 | OOM |
| ContextSpeech | 0.240 (x4.65) | 0.150 (x3.47) | 0.134 (x3.54) | 0.111 (x3.80) | 0.095 (x4.24) | 0.089 (x4.67) | 0.084 (x4.67) | 0.083 (x7.34) | 0.078 (x10.22) | 0.075 |

ContextSpeech as shown in Table 2). In summary, for extra-long input sentence, ContextSpeech model shows better performance in both inference time and memory tolerance, and also voice quality.

## 4.5 ABLATION STUDY

We conduct ablation study for the proposed ContextSpeech model to evaluate the effectiveness of the added module. Table 4 shows the paragraph CMOS results on Set-A for removing each component from ContextSpeech model.

**Memory Recurrence (MR)**. Memory reuse mechanism described in Section 3.1 is proposed to enlarge the receptive field of current segment to see more historical information. To verify the effectiveness of this module, we remove it from ContextSpeech model, and do a paragraph CMOS test for comparison. Set ContextSpeech model as 0 for reference, remove the MR component cause -0.085 regression, which demonstrates the contribution from MR mechanism.

**Text-based Contextual Encoder (TCE)**. As described in Section 3.2, we proposed a text-based contextual encoder to leverage hierarchical contextual information from plain text and broaden the model horizon to see more future information. To evaluate its effectiveness, we do paragraph CMOS test to compare the model with and without TCE module. The negative score -0.048 verifies the positive effect from TCE module.

**Linearized self-attention with Permute-based relative encoding (LP)**. LP is introduced in Section 3.3, which is used for improve model efficiency and capability especially on extra-long input. The efficiency improvement and corresponding performance on extra-long input are proved in Section 4.4. Here we replace the LP in ContextSpeech by previous softmax-based self-attention with relative position encoding in Transformer-XL to evaluate the performance in paragraphs with normal-length sentence. The paragraph CMOS result, -0.030, demonstrates that the LP module will not cause quality regression and even with slight improvement in normal case.

Table 4: Paragraph CMOS test result for Ablation Study.

| ContextSpeech | - MR | - TCE | - LP |
|---|---|---|---|
| 0 | -0.085 | -0.048 | -0.030 |

## 5 CONCLUSION

In this paper, we propose ContextSpeech, which is an expressive and efficient TTS model for paragraph reading. We introduce memory reuse mechanism in both encoder and decoder to transfer historical information of both text and speech to current sentence. Additionally, we encode text-based contextual information in a hierarchical structure to broaden the model horizon and enlarge the scope to see future information. Furthermore, we use linearized self-attention with permute-based relative position encoding to improve the model efficiency. Experiments on Chinese audiobook corpus demonstrate that ContextSpeech achieved superior voice quality and expressiveness in paragraph reading compared with one of the baseline model (76% reduction on the MOS gap to recording). ContextSpeech also alleviates the robustness issues in extra-short sentence (0.107 CMOS gain) and shows significant improvement in terms of the quality (0.226 CMOS gain) and efficiency (∼10x speedup) with the extra-long input.

**Future Work**. Besides long-form problem for paragraph in text, the audio content usually contains rich style and prosody variations like introducing the dialogue data in audiobook. There would be large potential for extracting more contextual representation from the speech data , for example, the memory reuse of prosody related information and appropriate style representation could bring richness improvement on the long-form content audio in further. We will continue explore this possibility in the future work.

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

## A    APPENDIX

**A. Objective Metrics**

We design a set of objective metrics to evaluate the prosody similarity between TTS wave and recording with the same transcripts, including pitch, intensity, duration and pause. Specifically, as shown in Figure 5, we use an internal vocoder analysis tool to generate the frame level f0 (fundamental frequency) and intensity of the given TTS wave and recording. It is worth mentioning that in order to get more accurate results, we perform linear interpolation and median filtering on f0 to obtain a continuous and smooth f0 curve. Given the transcription text, we can obtain the phoneme boundary, phoneme duration and pause duration between adjacent phonemes by an internal force align tool. In addition, we use an internal syllabification tool to obtain the syllable boundary of phoneme sequences.

Combing frame-level f0, intensity and the phoneme boundary, we can get phoneme-level f0 and intensity by averaging the frame-level features inside a phoneme. The duration of all the phonemes inside the syllable is added to get the duration of the syllable. The pause duration after the last phoneme of the syllable is treated as the pause duration of the syllable [2]. When the phone-level f0, intensity and syllable duration, pause features are ready, we calculate the pearson correlation and RMSE of the above prosody features between TTS wave and recording as the objective metrics. The pearson correlation is calculated by Eq.(7), where $x$ and $y$ are the prosody feature vector of TTS wave and recording respectively, $E_x$ and $E_y$ are the mean of the feature vector of $x$ and $y$. RMSE is calculated by Eq.(8), where $N$ is the vector length of $x$ and $y$. As we mentioned in section 4.1, these objective metrics are used as an additional reference combined with subjective evaluation for our model selection.

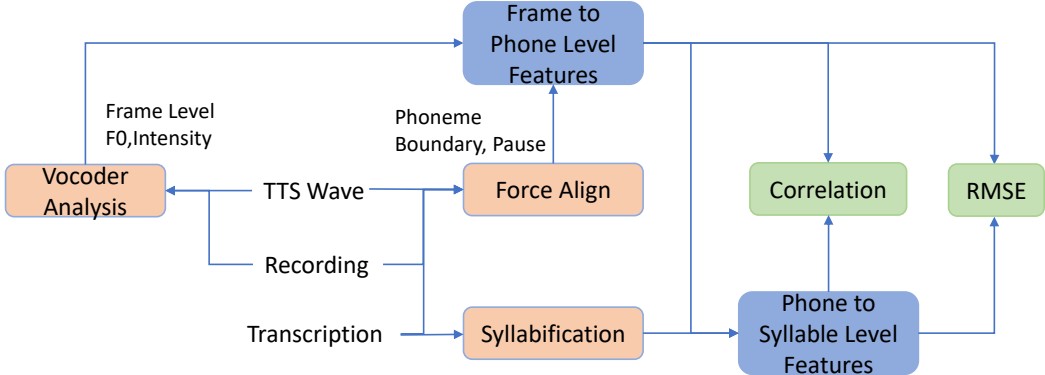

Figure 5: Objective Metrics.

$$pearson = \frac{\sum (x - E_x)(y - E_y)}{\sqrt{\sum (x - E_x)^2 \sum (y - E_y)^2}} \quad (7) \qquad rmse = \sqrt{\frac{\sum_{i=1}^{N}(x_i - y_i)^2}{N}} \qquad (8)$$

**B. Selection of Memory Length**

We have to select the best parameter setting of memory length for both encoder and decoder. As the number of different comparison pairs is large and the difference among these models is not obvious enough, we use the objective metrics described in Appendix A to perform the evaluation. Notice that the result here is based on the ContextSpeech model without the integration of text-based contextual encoder.

**Encoder Memory Length** To select the memory length of encoder, we first set the decoder memory length as 0, and then compare model performance under different encoder memory lengths (from

---
[2]From our experiments and observation, syllable-level features are more appropriate for the comparison of duration and pause related prosody feature.

32 to 512). The results are presented in Table 5. The distribution of the best metric values seems not concentrate on one model, since not all carried information in memory is useful for current sentence synthesis. We can see that the model with 128 as the encoder memory length achieves 3 best metrics, which is better than other models. 128 is also a number close to the average input length of 1 sentence in our data set. Therefore, we select 128 as the final setting of encoder memory length in our ContextSpeech model.

Table 5: Objective Metrics Result of Different Encoder Memory Length.

| Metrics | Correlation | | | | RMSE | | | |
|---|---|---|---|---|---|---|---|---|
| | Pitch | Intensity | Duration | Pause | Pitch | Intensity | Duration | Pause |
| 32-0 | 0.674 | 0.852 | 0.760 | **0.897** | 25.204 | 11.620 | 46.576 | **56.917** |
| 64-0 | 0.676 | 0.849 | 0.775 | 0.887 | 25.474 | 11.796 | 46.692 | 58.051 |
| 128-0 | **0.678** | 0.856 | **0.786** | 0.884 | **24.820** | 11.475 | 46.723 | 57.437 |
| 256-0 | 0.662 | 0.856 | 0.776 | 0.890 | 24.966 | **11.377** | 46.620 | 58.307 |
| 512-0 | 0.669 | **0.858** | 0.770 | 0.892 | 25.134 | 11.477 | **45.958** | 58.904 |

**Decoder Memory Length**. Given the memory length of encoder is 128, we compare models with different memory length in the decoder, from 16 to 512, and the result are shown in Table 6. We can see that the model with decoder memory length as 64 achieves the best overall performance among the parameter tuning range [16, 32, 64, 128, 256, 512]. Thus, we select 64 as the decoder memory length in the final ContextSpeech model. The results also reveal that longer memory will not produce better model performance, which can be illustrated form two respects: 1) We just need a small piece of speech at the end of last sentence to help the current sentence synthesis to the greatest degree. 2) Directly leveraging speech information by concatenation with previous decoder memory cannot make the most use of it, we need a more complex module to handle longer decoder memory. We will do more investigation on these observations in the future work.

Table 6: Objective Metrics Result of Different Decoder Memory Length.

| Metrics | Correlation | | | | RMSE | | | |
|---|---|---|---|---|---|---|---|---|
| | Pitch | Intensity | Duration | Pause | Pitch | Intensity | Duration | Pause |
| 128-16 | 0.674 | 0.843 | 0.740 | 0.889 | 25.973 | 11.969 | 48.107 | 58.418 |
| 128-32 | 0.686 | **0.849** | 0.759 | 0.891 | **24.920** | 11.777 | 47.862 | 57.390 |
| 128-64 | **0.707** | **0.849** | **0.786** | **0.894** | 24.980 | **11.562** | **45.693** | **55.876** |
| 128-128 | 0.666 | 0.845 | 0.761 | 0.890 | 25.476 | 11.821 | 48.118 | 58.084 |
| 128-256 | 0.664 | 0.846 | 0.763 | 0.891 | 26.703 | 11.835 | 47.421 | 57.824 |
| 128-512 | 0.677 | 0.843 | 0.761 | 0.891 | 25.677 | 11.906 | 47.227 | 57.961 |

## C. Definition of Metrics in Paragraph MOS

As we mentioned in paper, in the paragraph MOS, 25 native speakers listen to each audio and give a score in 10-point scale according to the overall performance and each specific metric, including naturalness, pleasantness, speech pause, stress, intonation, emotion, style matchiness and listening effort. The definition of each metric can be found in the following items. The price of this test is 0.1 dollar per case per judge.

- **Overall impression**. How is your overall impression on this content reading, considering the inside and cross sentences? Consider if the voice is clear, natural, expressive, easy to understand and pleasant to listen to.

- **Naturalness**. How nature is this content reading, considering the inside and cross sentence prosody?

- **Pleasantness**. If the voice sounds comfortable and pleasant reading this content

- **Speech pause**. If the break between words and the silence between sentences are appropriate?

- **Stress**. If the degree of emphasis is natural and correct?

- **Intonation**. If the melody and variation in the pitch level fits the sentence?

- **Emotion**. If the emotion is expressive and suitable for the content?

- **Style matchiness**. How much is the voice suitable for reading the content from the speaking style?
- **Listening effort**. How easy is it to focus on this voice and get information?

**D. Samples**

https://contextspeech.github.io/demo/

**E. Detail Steps of Some Equations**

- **ConformerBlock**
  ConformerBlock in Eq.(1) is presented in Fig.1 and the process can also be formatted with the following equations:

$$H'_{t,n} = ConvM(H^n_t) + H^n_t \tag{9}$$

$$H''_{t,n} = MHSA(H'_{t,n}) + H'_{t,n} \tag{10}$$

$$ConformerBlock(H^n_t) = ConvFFN(H''_{t,n}) + H''_{t,n} \tag{11}$$

- **Linearized Self-Attention with Permute-based Relative Position Encoding**
  Combine permutation operation in Eq.(6) into linearized self-attention, Eq.(4-5) can be rewritten as Eq.(12-13)

$$\mathcal{A}(Q_i, K, V) = \Big( \sum_{j=1}^{L} (r_i P^i_B \phi(Q_i))^T (r^{-j} P^j_B \phi(K_j)) V_j \Big) / \Big( \sum_{j=1}^{L} (r_i P^i_B \phi(Q_i))^T (r^{-j} P^j_B \phi(K_j)) \Big) \tag{12}$$

$$= \Big( r_i P^i_B \phi(Q_i))^T \sum_{j=1}^{L} (r^{-j} P^j_B \phi(K_j)) V_j \Big) / \Big( (r_i P^i_B \phi(Q_i))^T \sum_{j=1}^{L} (r^{-j} P^j_B \phi(K_j)) \Big) \tag{13}$$

As the $r$ is set as 1 in our model setting, the Eq.(13) can be simplified as:

$$\mathcal{A}(Q_i, K, V) = \Big( P^i_B \phi(Q_i))^T \sum_{j=1}^{L} (P^j_B \phi(K_j)) V_j \Big) / \Big( (P^i_B \phi(Q_i))^T \sum_{j=1}^{L} (P^j_B \phi(K_j)) \Big) \tag{14}$$

