# OpenReview forum: "ContextSpeech: Expressive and Efficient Text-to-Speech for Paragraph Reading"
_ICLR.cc/2023/Conference — Submitted to ICLR 2023_

### Official Review · Reviewer_bwpL · 2022-10-23

**Confidence:** 4
**Correctness:** 4
**Technical Novelty And Significance:** 3
**Empirical Novelty And Significance:** Not applicable
**Recommendation:** 6

**Clarity, Quality, Novelty And Reproducibility:**

The paper is technically solid and well written.

Some minor typos:
1. section 3.2: we use a BERTDevlin et al. -> we use a BERT Devlin et al.
2. “character” (fullwidth) -> "character" (halfwidth)


**Strength And Weaknesses:**

ContextSpeech is a nice combination of engineering and research efforts. By making use of caching mechanism and linearized self-attention module with corresponding permute-based relative position encoding, ContextSpeech can capture previous information with reduced computation cost and memory consumption. For the context awareness, ContextSpeech uses BERT embedding combined with pre-defined statistical information from contextual scripts, which broadens the model receptive field.

ContextSpeech has improved the paragraph-level prosody expressiveness compared to the baseline model. It generates high quality speech samples in terms of naturalness, intonation, stress and style. Also, the adapted speech rate makes extra-short utterance sound much more natural and close to human reading.

**Summary Of The Paper:**

This paper proposes ContextSpeech, a text-to-speech (TTS) system that models the contextual information in a paragraph for coherence and expressiveness, which constructs text-based semantic information in a hierarchical structure. At the same time, ContextSpeech doesn't largely increase the computation or memory cost by introducing a memory-cached recurrence mechanism. ContextSpeech improves the paragraph-level speech quality and prosody expressiveness in terms of subjective and objective evaluation metrics.

**Summary Of The Review:**

ContextSpeech achieves good performance in paragraph reading, including audio quality and inference speed. And the paper is well written. However, in consideration of the fact that most of the modules and algorithms in the pipeline are from previous work, my rating is also adjusted.

---

> ### Author Response · Authors · 2022-11-18
> **Response to Reviewer bwpL**
>
> Dear Reviewer,
>
> Thank you so much for your reviews and acknowledge on our technical contribution and write-up. We have carefully addressed your identified typos. We have also proofread the manuscript to identify other typos.

---

### Official Review · Reviewer_uAF5 · 2022-10-24

**Confidence:** 3
**Correctness:** 3
**Technical Novelty And Significance:** 3
**Empirical Novelty And Significance:** 1
**Recommendation:** 3

**Clarity, Quality, Novelty And Reproducibility:**

The contributions are somewhat new. Aspects of the contributions exist in prior work. Some important details are missing.


**Strength And Weaknesses:**

Strength:
The topic this paper studies is very interesting. I definitely agree that the context learning of TTS is not yet well-developed, and making progress on context learning will shed light on the some promising directions for natural speech synthesizing.

Weaknesses:
In section 2.1, "But the expressiveness is limited due to teacher-student distillation",  FastSpeech 2 requires no distillation.

The vocoder is not presented, I am curious whether the vocoder has a similar recurrent mechanism like Transformer-XL over the decoding process? If not, how do you join the output audios together?

The backbone of the proposed network is not well-presented. Is this model based on FastSpeech, or FastSpeech 2? The main body of the paper and Figure 1 clearly state that this model is based on FastSpeech. However, in the second paragraph of section 4.1, the authors present that the variance adapter is included which indicates that this model is based on FastSpeech 2. I am very curious whether the baseline model also uses FastSpeech 2 for a fair comparison?

The autoregressive models such as Tacotron 2 should be included in the comparison. Despite that AR models often suffer from relatively slower inference speed, it is computationally cheaper and suitable for synthesizing very long text.

The novelty of the method is limited since the Transformer-XL based TTS model has been discussed before.

The baseline model is weak, some modern TTS models such as FastSpeech 2, GlowTTS, VITS should be included in the comparison.

Typo: in the second paragraph of section 4.1 and the first paragraph of section 4.4, "ContexSpeech" should be  "ContextSpeech".

Missing a most relevant reference:
L. Xue, F. K. Soong, S. Zhang and L. Xie, "ParaTTS: Learning Linguistic and Prosodic Cross-Sentence Information in Paragraph-Based TTS," in IEEE/ACM Transactions on Audio, Speech, and Language Processing, vol. 30, pp. 2854-2864, 2022.


**Summary Of The Paper:**

This paper studies the problem of synthesizing extra-long text inputs such as a paragraph-level inputs for TTS tasks. The proposed method relies on the Transformer-XL like approach to capture the inter-sentence information and integrates BERT-based context information for both training and synthesis. The authors further employ a linearized self-attention module to reduce the overall memory cost and computational cost. Experimental results on an internal dataset confirms the effectiveness and efficiency of the proposed method.

**Summary Of The Review:**

While the paper’s premise is interesting, the submission suffers from poor presentation quality. and I’m not entirely convinced by the authors' claims when considering that some modern TTS models are not included for comparison. I also have concerns with the novelty of this work since the Transformer-XL based models and BERT-based context learning have been discussed from the prior works.

---

> ### Author Response · Authors · 2022-11-18
> **Response to Reviewer uAF5**
>
> Dear Reviewer,
>
> Thank you for your suggestion and feedback. Please find the response to your questions below.
>
> - Q1: Clarification of our baseline model.
>
>   R1: Our baseline model is ConformerTTS, which is a FastSpeech2-like model with variance adapter and it is also trained directly without the teacher-student distillation mode. We made some description about the baseline model in the first paragraph of Section 3.1. Due to space limit, we focus on the key contribution of contextual modelling in this work and ignore some details of our backbone model. We will add more technique description of our backbone method in our revised version.
> - Q2: Clarification of our used vocoder.
>
>   R2: We use MelGAN based vocoder in our experiments. We didn’t do any modification on the vocoder and the output audios will be concatenated directly.
> - Q3: Extension to integrate with other baseline models.
>
>   R3: Our proposed ContextSpeech framework is flexible to integrate with existing backbones, such as FastSpeech2, GlowTTS, and VITS. We use ConformerTTS as our backbone model. ConformerTTS is a FastSpeech2-like model with an improved Conformer block, which has been demonstrated to be effective, robust and efficient.
> - Q4: Discussion of ParaTTS.
>
>   R4: Thanks for pointing out this new published paper. ParaTTS has similar task goal as our work. The main contribution of ParaTTS is to leverage paragraph-level information from linguistics and prosody aspects, which is like an extension version of our text-based contextual encoder with additional prosody encoding. We have cited some representative contextual encoder relevant papers in related work section, and we will add this paper in our revised manuscript, thanks for reminder.

---

> > ### Comment · Reviewer_uAF5 · 2022-11-23
> > **Reply to Response to Reviewer uAF5**
> >
> > I have carefully read the rebuttal as well as other reviewers' opinions and the response to them. Unfortunately, none of my concerns are properly addressed：
> >
> > 1.The Clarification of our baseline model:
> >
> > Throughout the whole paper, there are many parts clearly state that both this model and ConformerTTS are based on FastSpeech, no corrections have been made during the first rebuttal period.
> >
> > 2: Clarification of our used vocoder.
> >
> > Thanks for the clarification. I'm glad to know that the vocoder is MelGAN. But unfortunately, throughout the paper, I haven't seen the presentation of vocoder included. And, why MelGAN?
> >
> > 3. Extension to integrate with other baseline models.
> >
> > The authors claim that this method can be integrated into other TTS models. However, the generalization is not guaranteed from careful studies. I totally agree that this work makes modifications to ConformerTTS, and comparisons between this work and ConformerTTS are essential. However, you need to evaluate this work and compare it with SOTA, and demonstrate the generalization to other TTS frameworks.
> >
> > 4. Discussion of ParaTTS.
> >
> > Apparently, ParaTTS should be included in the evaluations, as well as the comparisons to autoregressive models (eg. Tacotron2), SOTA TTS models (eg. VITS, NaturalSpeech), prior works on context modeling. Unfortunately, All these comparisons are omitted.
> >
> > I could see that the authors have put some effort in writing the rebuttal. I acknowledge that. But, unfortunately, in its current form the paper is not ready for a full publication.

---

> > > ### Author Response · Authors · 2022-12-10
> > > **Reply to Reviewer uAF5**
> > >
> > > - Clarification of our used vocoder.
> > >
> > >   In general Fastspeech like acoustic model + MelGAN vocoder is state-of-the-art TTS system, especially for industry speech product. In this paper, we mainly focus on the improvement on acoustic model. It is flexible to select any other vocoders (e.g., HifiGAN, WaveGAN, DiffWave), since the selection of vocoder is not the key problem.
> > >
> > > - Extension to integrate with other baseline models.
> > >
> > >   Actually, ConformerTTS is one representative TTS system which provides state-of-the-art speech synthesis performance. Thus, we use it as the baseline, and the performance improvement of our ContextSpeech model over the ConformerTTS can justify the effectiveness of our newly proposed method. We are writing this paper to provide some insights on better paragraph-level speech synthesis by sharing our proposed framework and essential experiment analysis. The extension to other various TTS model can be done depending on the specific demand, since the base model framework is not the key point to study in this paper.
> > >
> > >   The proposal and modification could be applied to autoregressive model (like s-Transformer) and Fastspeech-like acoustic model in this paper. In theory, it would work for VITS or NaturalSpeech too. However, due to the End-to-End modelling structure, the sample-level sequence is even challenging for directly long-form/paragraph modelling. There are several different strategies to apply it to e2e model. It would be a good topic for another paper to discuss rather than just in one paper.
> > >
> > >
> > > Thanks for your reviews and comments.

---

> > > > ### Comment · Reviewer_uAF5 · 2022-12-10
> > > > **Reply**
> > > >
> > > > I disagree with the statement that FastSpeech like acoustic model + MelGAN vocoder is state-of-the-art TTS system. End-to-end models such as VITS and NaturalSpeech that integrate the HiFi-GAN vocoder are more popular from my point of view. I agree that the choice of vocoder is not the key part but it is a necessary part that has to be clearly clarified.
> > > >
> > > > I disagree with the statement that the base model framework is not the key point to study in this paper. It is unclear whether the main insights of the work can be extended to other TTS systems, which is important for evaluating the value of this work.
> > > >
> > > >  'it would work for ..., it could be applied to ...,' As a scientific paper, these statements should be carefully studied, not by subjective judgment.

---

### Official Review · Reviewer_Wz2n · 2022-10-24

**Confidence:** 4
**Correctness:** 2
**Technical Novelty And Significance:** 1
**Empirical Novelty And Significance:** 1
**Recommendation:** 3

**Clarity, Quality, Novelty And Reproducibility:**

It requires substantial improvement for readability and clarity of the proposed method. Novelty also seems limited considering this is an ensemble of existing techniques such as efficient attention. Reproducing the experiments would be hard given the lack of details and clarity.


**Strength And Weaknesses:**

Strengths
- This paper tackles an important problem that has not received enough attention yet.
- The proposed method outperforms a simple baseline that do not have previous sentence as input

Weakness
- Presentation of this paper requires improvement. The current draft is barely readable
  - `\citep` should be used instead of `\citet` in most of the places to improve readability.
  - the diagram is not very clear or useful. It is unclear what Fig 1 (b) corresponds to in Fig 1 (a), and how “paragraph text” is different from “previous segment”. It is also unclear why there is a connection from “previous segment” to “current segment” in Fig 1 (b).
  - “where $t$ is the index of current sequence” is ambiguous. Does $t$ index tokens within a segment, or does it index segments?
  - It is unclear where the previous segment is used in Eq 1, which reads like a skip connection instead.
  - Fig 2 is too complicated and again unclear where it refers to in Fig 1 (a) / (b). How are F0-F5 used? What does “F0 = $i_{t_s}$ / $n_{t_s}$ = the i-th token / the number tokens in a sentence” mean?
  - Does “concat” and “Average Pooling” operate on $[E_{t-c}, \cdots E_{t+c}]$ or just E_t?
- This paper does not compare with any prior work that addresses context modeling, such as Wang et al. (2020). Instead, it only compares  with a model that does not have context from previous segments as input. The experiments are not sufficient


**Summary Of The Paper:**

This paper presents ContextSpeech, a text-to-speech synthesis model that additionally takes the embedding of the previous sentence as input to improve prosody modeling for paragraph reading. The authors also utilize efficient attention to improve efficiency.

**Summary Of The Review:**

The proposed methods are not described clearly. Novelty is limited. Experiments are insufficient.

---

> ### Author Response · Authors · 2022-11-18
> **Response to Reviewer Wz2n**
>
> Dear Reviewer,
>
> Thanks for your comments. We make the following further clarifications in response to your questions. We will carefully make modifications in our revised manuscript. We really appreciate your kind consideration to raise your score.
>
> - Q1: Description of Fig 1 (b)
>
>   R1: The title of Fig 1(b) “Conformer Block” has already been presented in the paper. Besides, the Conformer Block has also been highlighted and illustrated in Fig 1(a) with the same colour.
> - Q2: Clarification of “Paragraph Text” and “Current Segment”.
>
>   R2:
>
>   i) “Paragraph Text” is paragraph text sent to text-based contextual encoder. Since text is given for TTS system, we can leverage the paragraph-level text-based information.
>
>   ii) A paragraph can be partitioned into several segments, the TTS system will process them one by one. Thus, the “Current Segment” is the segment to be synthesised currently. The information from the “Previous Segment” will be leverage in our system to help improve the synthesis quality. The detail has already been described in Section 3.1 of Segment-level Memory Reuse.
> - Q3: Clarification of t.
>
>   R3: We have already defined “t” in the first paragraph of Section 3.1 in the manuscript. In particular, t is the segment index.
> - Q4: Typo in Eq.1.
>
>   R4: SG(H_t^n) should be SG(H_(t-1)^(n+1)). We have fixed this typo.
> - Q5: Clarification of Fig.2.
>
>   R5: Fig 2 is the Text-based Contextual Encoder, which is represented with the same name in deep yellow block in Fig.1(a).
> - Q6: Usage of F0-F5.
>
>   R6: F0-F5 will be concatenated with each token embedding of current sentence.
> - Q7: Clarification of i_(t_s) and n_(t_s).
>
>   R7: i_(t_s) is the index of current token in the sentence. n_(t_s) is the total number of tokens in current sentence.
> - Q8: Clarification of E.
>
>   R8: E is sentence-level embedding averaged by token-level Bert embedding corresponding to this sentence. E_(t-c),…,E_(t+c) are sentence embeddings inside the contextual window. These sentence embeddings will go through GRU to get a context-aware state vector, which will be concatenated with current sentence embedding E_t.
> - Q9: Clarification of baseline model-ConformerTTS.
>
>   R9: Wang et al. (2020) leverage the Transformer-XL framework into TransformerTTS system [1], which is an autoregressive TTS framework. Our adopted baseline model, the ConformerTTS, has been demonstrated to be effective than TransformerTTS framework [2]. Specifically, ConformerTTS is a non-autoregressive model, which is more efficient and robust than TransformerTTS. Thus, we adopt ConformerTTS as our baseline model.
>
> [1] Li N, Liu S, Liu Y, et al. Neural speech synthesis with transformer network[C]//Proceedings of the AAAI Conference on Artificial Intelligence. 2019, 33(01): 6706-6713.
>
> [2] Liu Y, Xu Z, Wang G, et al. Delightfultts: The microsoft speech synthesis system for blizzard challenge 2021[J]. arXiv preprint arXiv:2110.12612, 2021.

---

> > ### Comment · Reviewer_Wz2n · 2022-12-02
> > **Thanks for the response**
> >
> > I thank the authors for the detailed response. There are still a few places that are unclear
> >
> > >R4: SG(H_t^n) should be SG(H_(t-1)^(n+1)). We have fixed this typo.
> >
> > The notation is still incorrect. Eq 1 shows that $H^{n+1}_t$ has length of $H^{n+1}_{t-1}$ plus $H^n_t$, which means it grows over layers.
> >
> > >R8: E is sentence-level embedding averaged by token-level Bert embedding corresponding to this sentence.
> > Are you sure $E_t$ is a $d$-dimensional vector instead of a $L$-by-$d$ matrix (where $d$ is the BERT feature dimension)? The upper path in Figure 2 shows that $E_t$ is concatenated with token-level features (F0-F5).
> >
> > >R9
> > The main issue is that this paper only compares with a baseline that does not use context at all, while there exists quite a few studies leveraging context to address similar problems. The same issue is also raised by uAF5
> >
> > I'm raising my score to 3 considering that a few modeling details have been clarified. However, this paper lacks important comparison with the literature, and still requires significant improvement on writing.

---

### Official Review · Reviewer_7rCy · 2022-11-07

**Confidence:** 4
**Correctness:** 3
**Technical Novelty And Significance:** 4
**Empirical Novelty And Significance:** 4
**Recommendation:** 8

**Clarity, Quality, Novelty And Reproducibility:**

The paper's contributions are novel and the experimental and ablation results are interesting. However, I would recommend that the authors trim things down a bit to improve the clarity and flow of the article.

**Strength And Weaknesses:**

The paper is well motivated as it's attempting to solve practical problem of speech synthesis for highly phonetically contextual languages. The architectural changes are intuitive and the experimental results in terms of improved synthesis and reduced computational complexity support the changes.

The main weakness is the writing style as it gets winding sometimes. There are several repetitions of the same concept, perhaps to remind the readers about the authors' contributions but it doesn't detract from the technical contribution of the paper. Also, using pairwise comparison (PMOS) has been shown to be a better comparison for speech synthesis models than just MOS. It might also be good if another phonetically contextual languages is evaluated. English might be a good choice given the availability of datasets although it might not be as phonetically contextual as Chinese.

**Summary Of The Paper:**

The paper proposed a solution to the practical problem of contextual speech synthesis to address TTS model limitation for highly phonetically contextual languages such as Chinese. The paper employs the use of conformer architecture baseline and introduces 3 main changes to enable paragraph-level expressiveness - segment-level memory reuse using RNN based on Transformer-XL architecture,  text-based contextual embedding for both text and statistical features, as well as linearized self-attention to reduce transformer computational complexity. For the contextual text embedding, the context incorporates both the previous and future sentences in order to improve synthesis quality. Experimental results shows that the proposed contextual TTS performs better than the ConformerTTS baseline in terms of MOS score and performs close to the original recording as judged by human evaluators. Analysis also shows that the proposed architecture shows significant MOS improvement for paragraphs with both extra- long and short sentences with the extra-long sentences showing better improvement. The use of of inter-sentence memory reuse is also shown to significantly reduce the latency by 4x - 10x. Ablation results also show that each of the additional changes to the conformer TTS baseline contributes to the improved synthesis quality.

**Summary Of The Review:**

The paper addressed a practical problem facing speech synthesis for highly phonetically contextual languages such as Chinese. The proposed architectural changes are intuitive and novel, and the experimental and ablation results supports the proposed changes.

---

> ### Author Response · Authors · 2022-11-18
> **Response to Reviewer 7rCy**
>
> Dear Reviewer,
>
> Thank you for your constructive comments. We detail our response as below.
>
> - Q1: pairwise comparison (PMOS) for speech synthesis.
>
>    R1: Thanks for this suggestion. In our model evaluation, a similar evaluation method: CMOS (comparative mean opinion score) is adopted to evaluate the performance of our proposed ContextSpeech model (as shown in Table 2). Specifically, CMOS is very similar with PMOS with the consideration of pairwise samples from different models.
>
> - Q2: evaluation on another phonetically contextual language.
>
>    R2: Yes, we also did some experiments on English corpus as another phonetically contextual language. From the results, we summarize three key observations corresponding to our model design motivations:
>
>    i) The proposed contextual encoder achieves 0.12 paragraph COMS gain and 0.14 paragraph MOS gain.
>
>    ii) The memory reuse framework brings benefits to the model performance by achieving 0.415 CMOS gain in dealing with extra-long sentences.
>
>    iii) The linearized self-attention brings similar benefits in time and memory saving as Chinese corpus and achieves the same voice quality (0.01 MOS gap between linearized attention based model and softmax attention based model).

---

### Decision · Program_Chairs · 2023-01-20

**Decision:**

Reject

**Justification For Why Not Higher Score:**

The empirical evaluation needs to be strengthened. Minimally, this would be done by including a state-of-the-art TTS model such as VITS in the evaluation. Ideally, the authors would apply their ideas to a second TTS model that is closer to the mainstream to show that their ideas generalize beyond ConformerTTS.

The writing of the paper needs to be improved. The formatting of the citations should have been corrected after the reviewers mentioned it, the paper should be clear that ConformerTTS is based on FastSpeech2 (not FastSpeech), and the paper should specify which vocoder is used.

The experimental results should provide variances on the MOS scores.


**Justification For Why Not Lower Score:**

N/A

**Metareview: Summary, Strengths And Weaknesses:**

# Summary
This paper is focused on improving the quality of speech synthesis at the paragraph level. It is well known that synthesizing paragraphs sentence-by-sentence does not lead to acceptable results due to inconsistent expressiveness in the synthesis of successive sentences, missing paragraph level context leading to poor renditions of prosody, and trouble synthesizing especially short or especially long sentences. This paper attempts to improve paragraph-scale TTS by providing a segment-level recurrent memory (similar to Transformer-XL) in the encoder and decoder, which provides previous segment context to shape the synthesis of the current segment, and also text encodings that capture characteristics of the surrounding sentence context. To reduce the computational costs of introducing longer context into the synthesis process, a linearized self-attention mechanism with a permutation-based relative positional encoding are used. Experiments on a Chinese audiobook collection illustrate the advantages of the proposed approach.

# Strengths
- The paper is addressing an important research question, and the approach explored in the paper is well motivated.
- The engineering optimizations used in the paper (e.g., linearized self-attention and the specific relative positional encoding) are potentially useful in other TTS systems.

# Weaknesses
- VITS, as the current state-of-the-art in TTS, should have been included in the evaluation for comparison. Because ConformerTTS is less well known, it is difficult for readers to judge the significance of the improvements over that particular baseline.
- The writing needs to be improved: ConformerTTS is based on FastSpeech2, but the paper repeatedly says "FastSpeech," the vocoder used in the tested TTS system is never specified, and the formatting of the citations makes it difficult to read sections of the paper with many references.


**Summary Of Ac-Reviewer Meeting:**

# Summary of reviewer and AC live discussion of ICLR 2023 submission 2388

# Attendees:
- Area chair NraW
- Reviewer uAF5
- Reviewer bwpL
- Reviewer 7rCy

(Reviewer Wz2n was traveling and unable to attend, but shared their views via OpenReview and responded to a question from the AC)

# Area chair NraW summary
The level of agreement between the reviewers is greater than might be expected from the scores they have assigned to this paper.

All reviewers agree that the question of how to use paragraph-level context to improve the production of long-form synthetic speech is an important research question and that the approach explored in the paper is well-motivated. The reviewers also generally think that the engineering optimizations described in the paper are a solid contribution that would likely be useful to other practitioners.

The main point of disagreement is on the sufficiency of the empirical evaluation, with the more negative reviewers (uAF5, Wz2n, and bwpL) arguing that at the very least, the evaluation should have also included comparison to VITS as the current state of the art in TTS. This is particularly important given that the ConformerTTS baseline model is not one that is well known.

Some reviewers suggested that a comparison to another method for exploiting long-range context should have been included in the evaluation, but the AC has found so far that all papers mentioned by the reviewers are either concurrent work (Xue et al., 2022 and Makarov et al., 2022) or do not provide code (papers by Oplustil-Gallegos et al.), and has therefore concluded that is is not reasonable to ask for empirical comparisons against those papers.

Another concern shared by multiple reviewers is that there are problems with the writing of the paper. The ConformerTTS model is described as being based on FastSpeech, but the paper makes reference to components that are present only in FastSpeech2 such as the variance adapter, and the authors describe the model as being based on FastSpeech2 in the discussion, but these issues were not corrected in a revision. Similarly, the paper does not state which vocoder is used, although it emerged in the discussion between the reviewers and authors that MelGAN was used. Finally, a consistent formatting problem with the paper (use of \citet instead of \citep, which badly degrades readability) was not corrected in a revision even though it was raised by a reviewer.

# Reviewer uAF5 comments
- In the discussion the authors say they use a MelGAN vocoder, but this is not mentioned at all in the paper. Also, MelGAN is not as good as HiFi-GAN, meaning that the baseline is weaker than it ought to be.
- The proposed ConformerTTS baseline is derived from FastSpeech2, but the paper says FastSpeech.
- The empirical evaluation should include a SOTA baseline like VITS.

# Reviewer bwpL comments:
- The audio demonstrations are extremely impressive.
- The paper contains a nice combination of research and solid engineering work.
- These two factors led to the initial rating of 6.
- But, after the rebuttal, reading other reviews, and the discussion, agrees that a VITS baseline should be included in the empirical evaluation and finds that the writing of the paper is a bit rough.
- Because a big contribution of the paper is the engineering optimizations, the paper would be strengthened by a demonstration that they can be applied to another backbone. [AC - and, if the backbone were a more standard one and these were included in the empirical evaluation, the paper would be further strengthened.]

# Reviewer 7rCy comments:
- Interesting problem and a well-motivated approach.
- Would have preferred PMOS over MOS [AC - though the authors replied that they included CMOS results in the paper]
- Liked the optimizations in the paper a lot, thinks they would be extremely valuable to other researchers.
- Satisfied with the use of ablations to verify the proposed modeling approach, but can see the argument made by the other reviewers for a stronger baseline's being included in the empirical evaluation.